# OmniVCus: Feedforward Subject-driven Video Customization with Multimodal Control Conditions

**Yuanhao Cai**[1], **He Zhang**[2], **Xi Chen**[3,*], **Jinbo Xing**[4],
**Yiwei Hu**[2], **Yuqian Zhou**[2], **Kai Zhang**[2], **Zhifei Zhang**[2], **Soo Ye Kim**[2],
**Tianyu Wang**[2], **Yulun Zhang**[5,*], **Xiaokang Yang**[5], **Zhe Lin** [2], **Alan Yuille**[1]
[1] Johns Hopkins University, [2] Adobe Research, [3] The University of Hong Kong,
[4] The Chinese University of Hong Kong, [5] Shanghai Jiao Tong University

## Abstract

Existing feedforward subject-driven video customization methods mainly study single-subject scenarios due to the difficulty of constructing multi-subject training data pairs. Another challenging problem that how to use the signals such as depth, mask, camera, and text prompts to control and edit the subject in the customized video is still less explored. In this paper, we first propose a data construction pipeline, VideoCus-Factory, to produce training data pairs for multi-subject customization from raw videos without labels and control signals such as depth-to-video and mask-to-video pairs. Based on our constructed data, we develop an Image-Video Transfer Mixed (IVTM) training with image editing data to enable instructive editing for the subject in the customized video. Then we propose a diffusion Transformer framework, OmniVCus, with two embedding mechanisms, Lottery Embedding (LE) and Temporally Aligned Embedding (TAE). LE enables inference with more subjects by using the training subjects to activate more frame embeddings. TAE encourages the generation process to extract guidance from temporally aligned control signals by assigning the same frame embeddings to the control and noise tokens. Experiments demonstrate that our method significantly surpasses state-of-the-art methods in both quantitative and qualitative evaluations. Project page is at https://caiyuanhao1998.github.io/project/OmniVCus/

## 1 Introduction

Text-to-video diffusion generation models [1–3] have achieved great success in creating high-quality videos from user-provided text prompts. These advancements spark increasing interest in subject-driven video customization that aims to create a video for specific identities in user-provided images.

Current subject-driven video customization methods are mainly divided into two categories: tuning-based and feedforward methods. Tuning-based solutions [4–6] are time-consuming. They fine-tune adapters [7, 8]/LoRAs [9] attached to a pre-trained video diffusion model each time for one inference. In contrast, feedforward methods [10–14] integrate the visual embeddings of subjects into diffusion models during training in a data-driven manner to enable video customization without test-time tuning. Despite progress on feedforward methods, there are still some challenges as follows:

**(i)** Existing methods mainly study single-subject customization due to the difficulty of constructing multi-subject data pairs. Some works have explored multi-subject data construction but still have limitations. For instance, ConceptMaster [11] constructs closed-set data pairs with limited subject categories, which degrades the model's generalization ability. **(ii)** As the subjects in each training video are always limited, how to enable inference with more subjects is important but under-explored. **(iii)** How to add control conditions of different modalities to subject-driven customization is also less studied. These conditions include textual instructions to edit the subject in the generated video, camera trajectory to move the viewpoint, segmentation mask sequence, depth map, and so on. The data-preparation pipelines of previous methods also neglect to produce control signals for the subjects.

---

*Corresponding Authors

39th Conference on Neural Information Processing Systems (NeurIPS 2025).

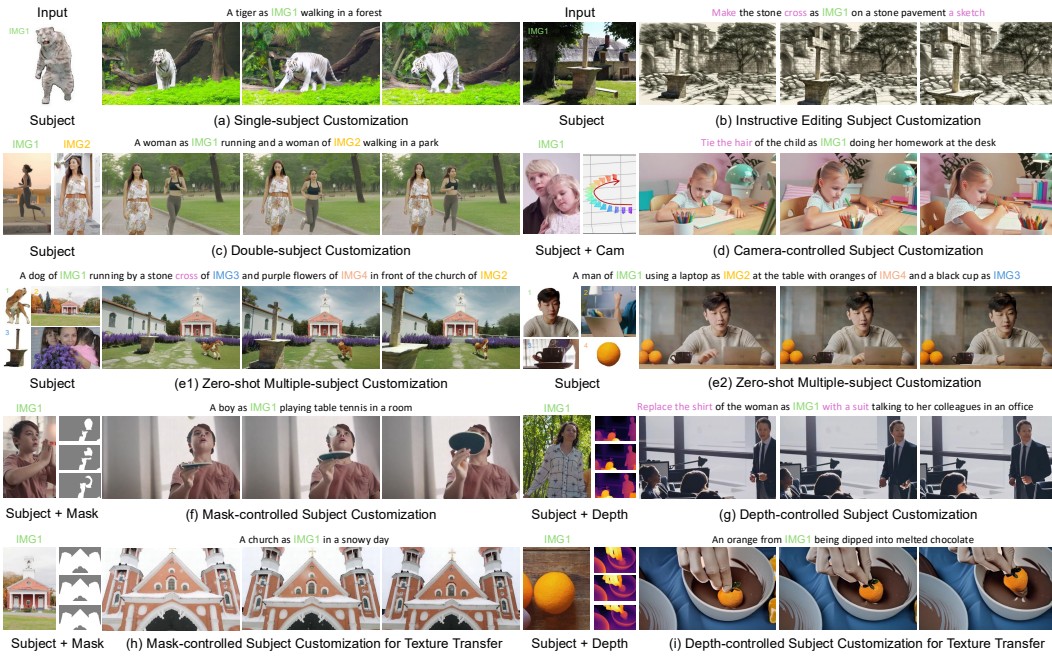

Figure 1: In (a) and (c), our method can change the pose and action of subjects. (b) The instructive editing texts are purple. In (e1) and (e2), our method trained with two subjects but can compose more subjects in inference. (d), (f), and (g) are the results under different controls. In the challenging cases of (g), (h), and (i), the subjects are unaligned with the mask or depth, but our method can adapt the shape or transfer the texture of the subjects.

To cope with these problems, we firstly propose a data construction pipeline, VideoCus-Factory, to produce training data pairs for multi-subject customization from raw videos without any labels. Our VideoCus-Factory first selects a frame from a video and uses a multimodal large language model to caption the frame and detect the subjects. Then we perform subject filtering, data augmentation, and random background placement to prevent the leakage of subject size, position, and background. This improves the variation and grounding ability of the training model and enables the inference of subject images with background. Based on our constructed data, we develop an Image-Video Transfer Mixed (IVTM) training with image instructive editing data to enable instructive editing effect for the subject in the customized video. Besides, our VideoCus-Factory can also generate signal data pairs such as depth-to-video and mask-to-video to control the subject-driven customization. Secondly, we propose a DiT-based framework, OmniVCus, to train on our constructed data. The input images, videos, and control signals are patchified, encoded, and concatenated into a long 1D token to input into OmniVCus. In particular, we design two embeddings in OmniVCus. As the number of subjects in a training video is limited, we propose a Lottery Embedding (LE) to enable customization with more subjects in inference than those used in training. The core idea of LE is to use a limited number of training subjects to activate the frame embeddings of more subjects. To enable more effective control effect of conditions, we propose a Temporally Aligned Embeddings (TAE). Our TAE assigns the same frame embeddings to the noise tokens and temporally aligned control tokens with dense semantic information, such as mask and depth. For the sparse viewpoint control signal without semantic information, TAE feeds them into a multi-layer perceptron (MLP) and then add them to the noise tokens to reduce the token length and computational complexity. Benefit from the constructed data and proposed techniques, our method can flexibly compose multimodal conditions to control the video customization, as shown in Fig. 7. In a nutshell, our contributions can be summarized as:

**(i)** We design a data construction pipeline, VideoCus-Factory, to produce training data pairs and control signals for subject-driven video customization from only raw videos. Based on our data, we develop an IVTM training strategy to enable instructive editing effect of the subject in the video.

**(ii)** We propose a new method, OmniVCus, with two embedding designs, LE and TAE, for subject-driven video customization. LE enables video customization with more subjects in inference than training. TAE enables more effective control of temporally aligned signals to the customized video.

**(iii)** Experiments show that our method significantly outperforms state-of-the-art (SOTA) methods in quantitative metrics while yielding more visually favorable, editable, and controllable results.

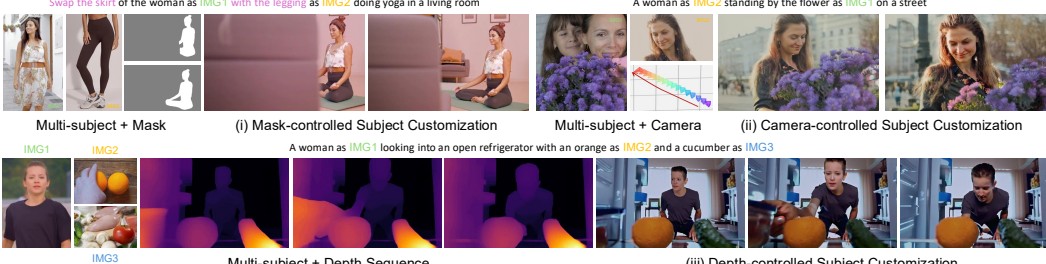

Figure 2: Our method can flexibly compose different conditions to control multi-subject video customization.

## 2 Related Work

**Text-to-Video (T2V) Diffusion Models** [1–3, 15–28] have witnessed significant progress in recent years. Preliminary T2V diffusion models are mainly based on stable diffusion [29], which formulates the diffusion process in latent space and uses a U-shaped convolutional neural network (CNN) [30] as the denoiser. Yet, the model capacity of CNN is limited for large-scale training. Thus, a later work DiT [31] employs Transformer [32] to replace U-Net in diffusion. These DiT-based methods [2,33–43] show very impressive performance in video generation and flexibility to add control conditions just by extending the input 1D tokens. This work exploits the advancement of the DiT framework to explore its potential in subject-driven video customization under different modalities of control signals.

**Subject-driven Video Customization** approaches are mainly divided into two categories: tuning-based [4,5,44–52] and feedforward methods [12,14,53–60]. Prior tuning-based methods are typically focused on single-subject scenarios. For instance, DreamVideo [4] fine-tunes an identity adapter and combines textual inversion to customize video for a subject. Videomage [52] employs subject and motion LoRAs to capture personalized content for multi-subject customization. These methods require a long time for inference. To avoid test-time tuning, later works [53] develop feedforward solutions. Some recent efforts [11, 12, 54] such as ConceptMaster [11] and Video Alchemist [12] try to construct data pairs for multiple subjects but their data pipelines still have limitations. Plus, how to add control to subject-driven video customization is still under-explored.

## 3 Method

### 3.1 VideoCus-Factory

Our data construction pipeline, VideoCus-Factory, is depicted in Fig. 3. VideoCus-Factory can produce training data pairs for multi-subject video customization from raw videos without any labels.

**Video Captioning.** For a video sequence, we first randomly select a frame and then use the multi-modal large language model, Kosmos-2 [61], to caption it and detect the subjects in the frame. Take the video in Fig. 3 as example, Kosmos-2 outputs the caption "An image of a bride and groom walking away from a car and looking back at it", a list of subjects ["a bride", "groom", "car"], the starting and ending positions of the subjects in the caption [[12, 19], [24, 29], [48, 53]], and bounding boxes of the detected subjects. We modify the caption by removing the prefix "An image of" and plug in image labels such as IMG1 and IMG2 corresponding to the subjects with their positions in the caption.

**Subject Filtering.** We feed the detected bboxes and raw video into SAM2 [62] to track and segment the subjects. Then we filter out the failure segmentation cases by thresholding the average segmentation values across frames. For example, in Fig. 3, "a car" is not segmented in some frames. Thus, we filter it out. We also filter out some word clouds and large background without identity.

**Data Augmentation.** If we directly use the images of segmented subjects to train a generation model, the scales, positions, and poses of subjects are leaked during training. As a result, the model tends to learn image animation with less variation. In addition, since the segmented subjects do not have background like ConceptMaster [11], directly training with this data degrades the model's grounding ability, limits its application as it needs to crop out the subject first, and easily leads to the copy-paste effect. To handle these problems, we randomly rotate the subjects, rescale them, move them to the center position, and augment their colors. Then we randomly select an image, which may also be pure white, and place it as the background for the subject to derive the input images of training data.

**Control Signals.** To add control conditions, VideoCus-Factory also constructs control signal training data pairs. As shown in Fig. 3, the mask sequence of the subject and the raw video with the modified

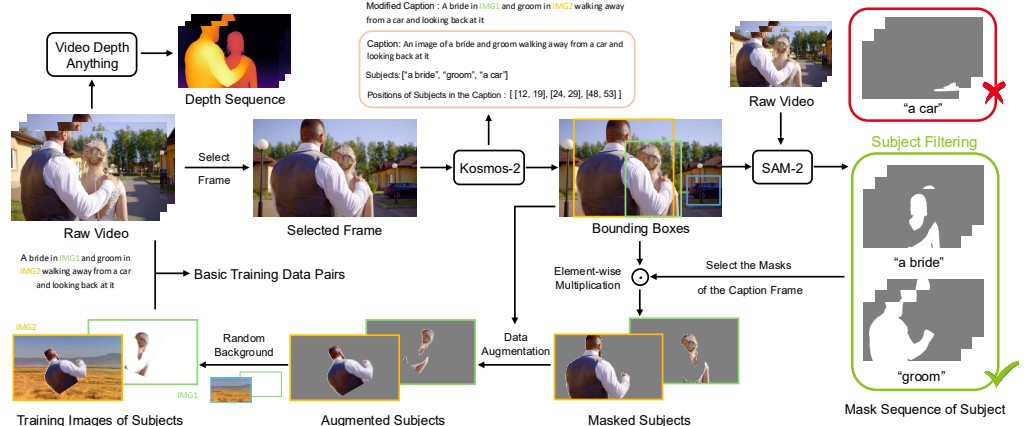

Figure 3: Our data construction pipeline VideoCus-Factory uses Kosmos-2 [61] to caption the raw video and detect the subjects. Then we use SAM-2 [62] to segment and filter the detected subjects to derive the training input images. VideoCus-Factory also constructs control data pairs such as mask-to-video and depth-to-video.

caption without image labels form the data pairs for mask-to-video control data. The depth sequence predicted by the video-depth-anything [63] and the raw video with the same caption form the depth-to-video control data pairs. Note that the mask-to-video and depth-to-video data are not paired with the subject-driven customization data in training. Our model can flexibly compose them in inference.

## 3.2 OmniVCus

As shown in Fig. 4, OmniVCus is a DiT-based framework. It can be mixed trained with different tasks, including single-/double-subject customization, depth-/mask-to-video, text-to-multiview, text-to-image/-video, and image instructive editing. Texts, images, and videos are patchified, encoded into the latent space, concatenated with the noise tokens, and fed into the full-attention DiT. OmniVCus can flexibly compose tokens of different signals to control and edit the subject in the customized video. We notice that the condition frame tokens are mainly divided into two parts: the image tokens containing the subjects, and the other temporally aligned tokens of control signals. To handle these two types of tokens, we design Lottery Embedding (LE) and Temporally Aligned Embedding (TAE).

**Lottery Embedding.** As the subjects in each training sample are limited, it is important to enable customization with more subjects in inference than training. To this end, our Lottery Embedding (LE) uses the limited subjects in the training samples to activate more frame embeddings, as shown in Fig. 4 (a). Denote the max number of subjects in a training sample as $K$ and the number of subjects we aim to compose as $M(M > K)$. Then LE randomly selects a set $\mathcal{S}$ of $K$ numbers from $[1, M]$ as

$$\mathcal{S} \sim \text{Unif}\big(\{\, \mathcal{A} \subseteq \{1, \ldots, M\} \mid |\mathcal{A}| = K \,\}\big), \tag{1}$$

where Unif denotes the uniform distribution. Since the Transformer is unordered, we need to create an order for it on the frame embedding and match the order of the image index label. Thus, we sort $\mathcal{S}$ in ascending order to derive $\mathcal{S}_\uparrow$ and then assign the elements of $\mathcal{S}_\uparrow$ as frame embeddings to the input images of subjects. These frame embeddings are reshaped, undergo an MLP, and then added to the tokens of the corresponding subject images. By our LE, we can activate more frame position embeddings during training and enable zero-shot more-subject video customization in inference.

**Temporally Aligned Embedding.** We notice that the control signals, such as camera, depth, and mask are temporally aligned with the generated video. Thus, to better direct the Transformer model to extract guidance from these control signals, TAE assigns the same frame embeddings to the control tokens and noise tokens. In Fig. 4 (b), we denote the length of the generated video as $N$. Then the frame position embeddings for control tokens and noise tokens are $\{M + 1, M + 2, \cdots, M + N\}$. To distinguish the control signals and noise, we only add the timestep embedding to the noise tokens.

In particular, the depth and mask sequences are structural signals containing fine-grained spatial and semantic information. Hence, we use a 3D-VAE to encode them to preserve these information. In contrast, the camera signals do not contain such information. On the other hand, the computational complexity of Transformer is quadratic to the length of input tokens. Thus, we integrate the camera signals into noise instead of concatenating them. Specifically, to enhance the control of camera signals and spatially align them to the noise tokens, we adopt the pixel-aligned ray embeddings,

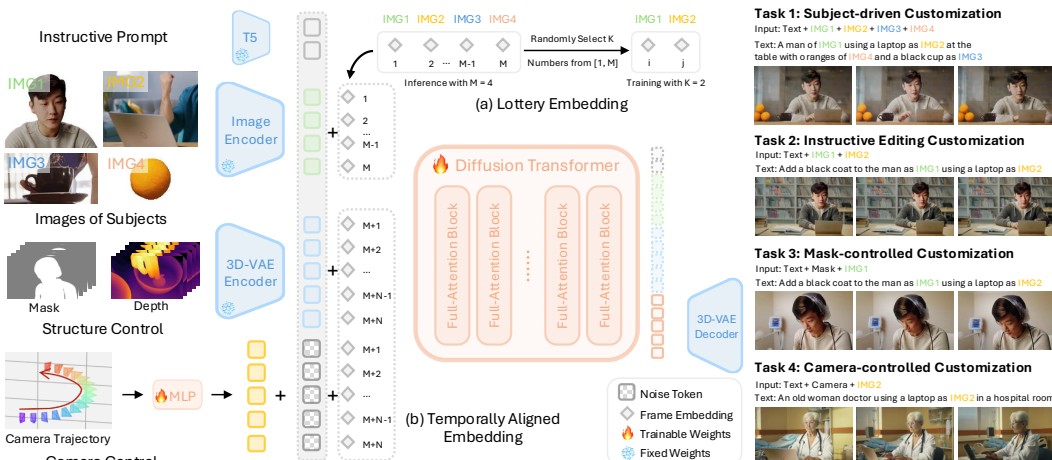

Figure 4: OmniVCus is DiT architecture that can compose different input signals to customize a video. (a) LE enables more-subject customization in inference by activating more frame embeddings with training subjects. (b) TAE extracts the guidance from control signals by aligning the frame embeddings of condition and noise tokens.

plücker coordinates, parameterized as $r = (o \times d, d)$, where $o$ and $d$ are the position and direction of the ray landing on a pixel. Then the plücker coordinates are patchified and undergo an MLP to add with the noise. Denote the input structure control tokens as $\mathbf{T}_{sc}^{in}$, the noise tokens as $\mathbf{T}_n^{in}$, and the tokens of plücker coordinates as $\mathbf{T}_c^{in}$. Then the mapping function of TAE, $f_{\text{TAE}}$, is formulated as

$$
\begin{aligned}
(\mathbf{T}_{sc}^{out}, \mathbf{T}_n^{out}) &= f_{\text{TAE}}(\mathbf{T}_{sc}^{in}, \mathbf{T}_n^{in}, \mathbf{T}_c^{in}) \\
&= \left( \mathbf{T}_{sc}^{in} + \text{MLP}_f(p_f), \mathbf{T}_n^{in} + \text{MLP}_f(p_f) + \text{MLP}_t(t) + \text{MLP}_c(\mathbf{T}_c^{in}) \right),
\end{aligned}
\tag{2}
$$

where $p_f$ is the frame position, $t$ is the timestep, and $\text{MLP}_f$, $\text{MLP}_t$, $\text{MLP}_c$ are three MLPs. The output tokens $\mathbf{T}_{sc}^{out}$ and $\mathbf{T}_n^{out}$ are then concatenated with other tokens to be fed into the DiT model.

**Image-Video Transfer Mixed Training.** Due to the lack of training data pairs for subject-driven customization with instructive editing. We develop an Image-Video Transfer Mixed (IVTM) training strategy to enable the editing effect for the subject of interest without constructing new data pairs. As the image instructive editing training data is sufficient, our goal is to transfer the editing effect from image to video. To this end, we need to construct a common task on image and video as the bridge for the knowledge transfer of instructive editing from image-to-image to image-to-video. In our IVTM training, this common task pair is single-subject image and video customization. For the image customization, we select the caption frame in Fig. 3 with the processed image of subject as the training pairs. In Fig. 4, we align the frame position embeddings of the input images of the image instructive editing and single-subject image/video customization in our IVTM training for better transferring. This frame embedding is also assigned by our LE in Eq. (1) with $K = 1$ to allow better composing instructive editing with multi-subject customization. In inference, we compose the prompts of image instructive editing and subject-driven customization to activate the editing effect.

**Training Objective.** Our model is mixed trained with different tasks using the flow-matching loss [64, 65]. Specifically, denoting the ground-truth video latents as $\mathbf{X}^1$ and noise as $\mathbf{X}^0 \sim \mathcal{N}(0, 1)$, the noisy input is produced by linear interpolation $\mathbf{X}^t = t\mathbf{X}^1 + (1 - t)\mathbf{X}^0$ at timestep $t$. The model $v_\theta$ predicts the velocity $\mathbf{V}^t = \frac{\partial}{\partial t}\mathbf{X}^t = \mathbf{X}^1 - \mathbf{X}^0$. Then the overall training objective is formulated as

$$
\min_\theta \mathbb{E}_{t, \mathbf{X}^0, \mathbf{X}^1} \left[ \left|\left| \mathbf{V}^t - v_\theta(\mathbf{X}^t, t \mid C_{txt}, C_{img}, C_{depth}, C_{mask}, C_{traj}) \right|\right|_2^2 \right],
\tag{3}
$$

where $C_{txt}, C_{img}, C_{depth}, C_{mask}$, and $C_{traj}$ denote the input texts, images, depths, masks, and cameras. As different training tasks are not paired with each other, some input conditions are missing in different training samples. But our model can flexibly compose different input signals in inference.

## 4 Experiment

### 4.1 Experimental Settings

**Dataset.** For subject-driven video customization, depth-to-video, and mask-to-video generation, we use our VideoCus-Factory to create ∼1.2M, ∼1.4M, and ∼1.6M training data pairs. The data for these

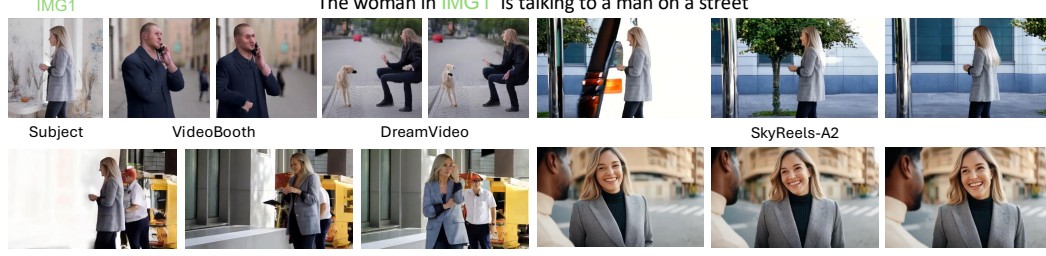

Figure 5: Visual comparison of single-subject video customization with state-of-the-art algorithms. Our method can change the pose and viewpoint of the subject while keeping the identity such as the hair, jacket, and sweater.

| Methods | Subject | CLIP-T | CLIP-I | DINO-I | Consistency | Dynamic |
|---|---|---|---|---|---|---|
| VideoBooth [53] | Single | 0.2541 | 0.5891 | 0.3033 | 0.9593 | 0.4287 |
| DreamVideo [4] | Single | 0.2799 | 0.6214 | 0.3792 | 0.9609 | 0.4696 |
| Wan2.1-I2V† [1] | Single | 0.2785 | 0.6319 | 0.4203 | 0.9754 | 0.5310 |
| SkyReels [54] | Single | 0.2820 | 0.6609 | 0.4612 | 0.9797 | 0.5238 |
| **Ours** | Single | 0.3293 | 0.7154 | 0.5215 | 0.9928 | 0.5541 |
| SkyReels [54] | Multiple | 0.2785 | 0.6429 | 0.4107 | 0.9710 | 0.5892 |
| **Ours** | Multiple | 0.3264 | 0.6672 | 0.4965 | 0.9908 | 0.6878 |

(a) Comparison of subject-driven video customization.

| Methods | Alignment | Identity | Quality |
|---|---|---|---|
| VideoBooth [53] | 91.9 | 97.3 | 94.6 |
| DreamVideo [4] | 89.2 | 89.2 | 97.3 |
| Wan2.1-I2V† [1] | 83.8 | 86.4 | 73.0 |
| SkyReels [54] | 75.7 | 81.1 | 70.3 |
| **Ours** | – | – | – |
| SkyReels [54] | 73.0 | 78.4 | 67.6 |
| **Ours** | – | – | – |

(b) User preference (%) of our method.

| Methods | CLIP-T | CLIP-I | DINO-I | Consistency | Dynamic |
|---|---|---|---|---|---|
| VideoBooth [53] | 0.2453 | 0.5935 | 0.2989 | 0.9570 | 0.5825 |
| DreamVideo [4] | 0.2642 | 0.6116 | 0.3511 | 0.9604 | 0.5760 |
| Wan2.1-I2V [1] | 0.2690 | 0.6208 | 0.3722 | 0.9734 | 0.6157 |
| SkyReels [54] | 0.2761 | 0.6368 | 0.4259 | 0.9635 | 0.5904 |
| **Ours** | 0.3126 | 0.7061 | 0.4942 | 0.9915 | 0.6226 |

(c) Comparison of instructive editing for subject customization.

| Methods | Alignment | Identity | Quality |
|---|---|---|---|
| VideoBooth [53] | 94.6 | 94.6 | 91.9 |
| DreamVideo [4] | 91.9 | 97.3 | 94.6 |
| Wan2.1-I2V [1] | 78.4 | 81.1 | 67.6 |
| SkyReels [54] | 73.0 | 75.7 | 64.9 |
| **Ours** | – | – | – |

(d) User preference (%) of our method.

| Methods | CLIP-T | CLIP-I | DINO-I | Consistency | Dynamic |
|---|---|---|---|---|---|
| Motionctrl [66] | 0.2984 | 0.5215 | 0.2066 | 0.9857 | 0.4272 |
| Cameractrl [67] | 0.2909 | 0.5163 | 0.1982 | 0.9711 | 0.5845 |
| CamI2V [68] | 0.2871 | 0.5365 | 0.2248 | 0.9660 | 0.5623 |
| **Ours** | 0.3104 | 0.6751 | 0.5233 | 0.9911 | 0.6204 |

(e) Comparison of cameral-controlled subject customization.

| Methods | Alignment | Identity | Quality |
|---|---|---|---|
| Motionctrl [66] | 91.9 | 86.8 | 78.4 |
| Cameractrl [67] | 70.3 | 89.2 | 91.9 |
| CamI2V [68] | 73.0 | 83.8 | 81.1 |
| **Ours** | – | – | – |

(f) User preference (%) of our method.

Table 1: Quantitative results and user study of state-of-the-art subject-driven video customization methods.

three tasks are not paired with each other, and every input video sequence to our VideoCus-Factory is randomly selected from our internal video data pool. For subject-driven video customization, the scale factor is randomly selected from 0.7 to 1.3 and the color augmentation includes brightness scaling (0.9∼1.1), linear contrast adjustment (0.9∼1.1), saturation scaling (0.9∼1.1), and hue shift (-10°∼10°). For text-to-multiview, we select 320K samples from Objaverse [69] labeled with long and short text prompts as the training samples. We adopt the OmniEdit [70] as the image instructive editing dataset containing 1.2M data pairs. Besides, we also fine-tune the model with text-to-image (∼300M) and text-to-video (∼1M) data. In evaluation, we collect 112 samples for single-subject customization and instructive editing customization, 76/74/56 samples for double-/triple-/quadruple-subject customization, and 112 samples for camera-controlled subject-driven video customization.

**Implementation Details.** Our model is fine-tuned from a T2V model with 5B parameters for 100K steps in total at a batch size of 356 on 64 A100 GPUs for 5 days. We adopt the AdamW optimizer [71] ($\beta_1 = 0.9$, $\beta_2 = 0.95$) with a weight decay of 0.1. The learning rate is linearly warmed up to $1e^{-5}$ with 2K iterations and decays to $1e^{-6}$ using cosine annealing [72]. The spatial resolution of training images and videos is set to $512 \times 512$ for text-to-multiview and image instructive editing and $384 \times 640$ for other tasks. The frame number and fps of the training video are set to 64 and 24. We use five metrics for evaluation. (1) CLIP-T computes the average cosine similarity between CLIP [73] image embeddings of all generated frames and their text embedding. We remove the image index label and adapt the text prompts after instructive editing when computing CLIP-T. (2) CLIP-I calculates the average cosine similarity between the CLIP image embeddings of all generated images and the target images. (3) DINO-I [74] also measures the visual similarity between the generated and target subjects using ViTS/16 DINO [75]. (4) Temporal Consistency computes the CLIP image embeddings

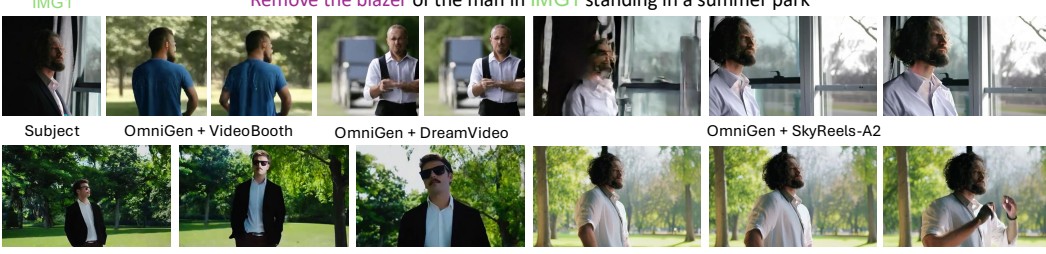

Figure 6: Visual comparison of instructive editing for subject-driven video customization with SOTA methods. VideoBooth, DreamVideo, and SkyReels-A2 first use OmniGen to edit the subject and then customize the video. Wan2.1-I2V [1] uses OmniGen to edit and customize the subject and then animate the image to derive the video.

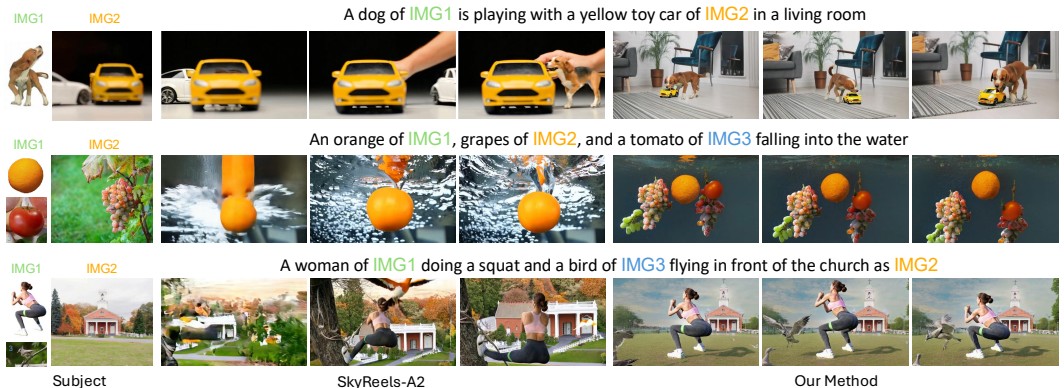

Figure 7: Visual comparison of multi-subject video customization with the SOTA method SkyReels-A2 [54].

and averages the cosine similarity between every pair of consecutive frames. (5) Dynamic Degree is computed as the optical flow predicted by RAFT [76] magnitude between consecutive frames.

## 4.2 Main Results

**Composing Different Control Conditions.** As shown in Fig. 1 and 7, our model can flexibly compose different control signals. **(i)** In Fig. 1 (b), benefit from IVTM training, our model can modify the subject and transfer its style to sketch. **(ii)** Benefit from LE, our model trained with 2 subjects but can compose 4 subjects in inference, as shown in Fig. 1 (e1) and (e2). **(iii)** In Fig. 1 (f) and (g), our model can change the pose and action of subjects following the mask or depth. (g) is a hard case where the depth is from a man but our model can fill the depth with the woman while keeping the face identity and swapping her shirt with a suit following the instruction. In harder cases where the subjects are severely unaligned with the mask or depth, our model can still transfer the texture of subjects. In Fig. 1 (h), the model transfers the appearance of church to the mask of house. In Fig. 1 (i), the texture of orange is transferred to the depth of strawberry. **(iv)** Even for more challenging multi-subject cases under different control signals in Fig. 7, our model can still robustly handle them.

**Comparison with SOTA Methods. (i)** We compare OmniVCus with 4 SOTA methods including an I2V method (Wan2.1-I2V [1]), two single-subject video customization methods (DreamVideo [4] and VideoBooth [53]), and a multi-subject video customization method (SkyReels-A2 [54]) on subject-driven video customization without and with instructive editing in Tab. 1a and 1c. In Tab. 1a, Wan2.1-I2V [1] uses the SOTA image customization model OmniGen [77] to first customize the subject and then animate it. The results for multi-subject customization are averaged on double- and triple-subject customization, as SkyReels-A2 can support at most three subjects. In 1c, OmniGen is first used to instructively edit the subject for all compared baselines as they show limitations in editing the subject itself. Our OmniVCus significantly outperforms previous methods in all tracks in both Tab. 1a and 1c, suggesting its advantages in identity preserving and high-quality video generation. Fig. 5, 6, and 7 show the visual comparisons of single-subject customization without and with instructive editing and multi-subject customization. In Fig. 5, our method can better change the pose of the woman while keeping the identity such as the hair, jacket, and sweater. In Fig. 6, our method can better follow the text to remove the blazer while keeping the man's identity and customize the background. In Fig. 7, SkyReels-A2 misses some subjects and tends to animate the image. In contrast, our method can better compose all the subjects and follow the texts to customize the video.

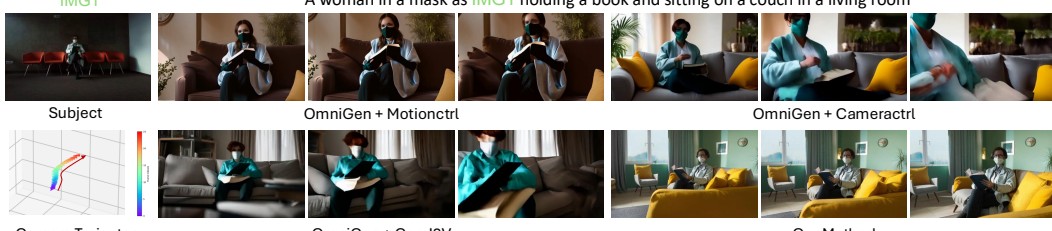

Figure 8: Comparison of cameral-controlled subject-driven video customization. Motionctrl, Cameractrl, and CamI2V first employ OmniGen to customize the image of subject and then animate it following the camera.

| Method | CLIP-T | DINO-I | Consistency | Dynamic |
|---|---|---|---|---|
| Baseline | 0.2175 | 0.2405 | 0.9588 | 0.3759 |
| + Subject Filtering | 0.2431 | 0.5053 | 0.9617 | 0.3826 |
| + Data Augementation | 0.3293 | 0.5215 | 0.9928 | 0.5541 |

(a) Ablation of VideoCus-Factory data construction

| Embedding | CLIP-T | DINO-I | Consistency | Dynamic |
|---|---|---|---|---|
| Naive | 0.2618 | 0.2947 | 0.9751 | 0.4948 |
| Add-to-Noise | 0.1722 | 0.1680 | 0.9319 | 0.5437 |
| Our TAE | 0.3054 | 0.3794 | 0.9909 | 0.4965 |

(b) Ablation of our Temporally Alignment Embedding

| Emebedding | CLIP-T | DINO-I | Consistency | Dynamic |
|---|---|---|---|---|
| w/o LE | 0.2105 | 0.3364 | 0.9702 | 0.6943 |
| with LE | 0.2728 | 0.4163 | 0.9810 | 0.6806 |

(c) Ablation study of our Lottery Embedding mechanism

| Training Method | No Mixed | Direct Mixed | Our IVTM |
|---|---|---|---|
| CLIP-T | 0.2137 | 0.2585 | 0.3126 |
| User Pref. (%) | 91.9 | 75.7 | — |

(d) Ablation of our mixed training for editing effect

Table 2: Ablation study. (a) is conducted on single-subject customization. (b) is done on depth-controlled customization. (c) is done on multi-subject customization. (d) is done on instructive editing subject customization.

**(ii)** Tab. 1e compares OmniVCus with three SOTA camera-controlled I2V methods (Motionctrl [66], Cameractrl [67], and CamI2V [68]). They use OmniGen to customize the subject first and then animate it. Our method surpasses the recent best method CamI2V by 0.2985 in DINO-I. Fig. 8 shows the visual results. Our method can better keep the identity of woman and follow the camera trajectory.

**(iii)** We also conduct a user study with 37 participants on single-subject video customization without and with instructive editing and camera-controlled subject-driven video customization in Tab. 1b, 1d, and 1f. Each participant views the videos generated by OmniVCus and a random competing method, along with the images of the subject, text prompts, and control signals. The participants are asked three questions: 1) Which video aligns better with the customization prompt/editing instruction/camera trajectory? 2) Which video better keeps the identity of the subject? 3) Which video has better quality? As reported in Tab. 1b, 1d, and 1f, reports the user preference (%) of our method over competing entries. Our method outperforms all SOTA methods by a large margin.

### 4.3 Ablation Study

**VideoCus-Factory.** We conduct experiments on single-subject video customization to study the steps of our VideoCus-Factory in Tab. 2a. We remove the subject filtering and data augmentation including random background placement from VideoCus-Factory as the baseline. When we apply the subject filtering, the DINO-I score significantly improves by 0.2648 because the left training video data after filtering can keep the subject in all frames. Subsequently, when using the data augmentation, the CLIP-T and dynamic degree gain by 0.0862 and 0.1715. This is because the segmented subject without data augmentation leaks the position, scale, and background in the training process. As a result, the customized videos have less variation and suffer from the copy-paste issue. As shown in Fig. 9 (a), with our data augmentation, the dog can change its pose in the customized video.

**Temporally Aligned Embedding.** We conduct experiments to study the effect on depth-controlled subject video customization in Tab. 2b. We compare our TAE with two options: 1) Naive method that assigns different frame embeddings to the tokens of depth (length $N$) and noise (length $N$) from $M + 1$ to $M + 2N$ and the timestep embedding is added to all tokens. 2) Adding the depth to the noise after undergoing an MLP. As listed in Tab. 2b, the add-to-noise method causes model collapse because the fine-grained spatial information in depth tokens is corrupted by the noise. Our TAE surpasses the naive embedding by a large margin in CLIP-T and DINO-I scores. As shown in Fig. 9 (c1) and (c2), our TAE can help the model better follow the guidance of depth to generate higher-quality video with fewer artifacts. Besides, TAE can help the control signals to be better composed with other tokens. The naive embedding fails to customize the bottle of the beer from text tokens and keep the identity of the boy from image tokens. In contrast, TAE can preserve the identity and follow the prompts to customize the scenario. Fig. 9 (d1) and (d2) compare the naive embedding

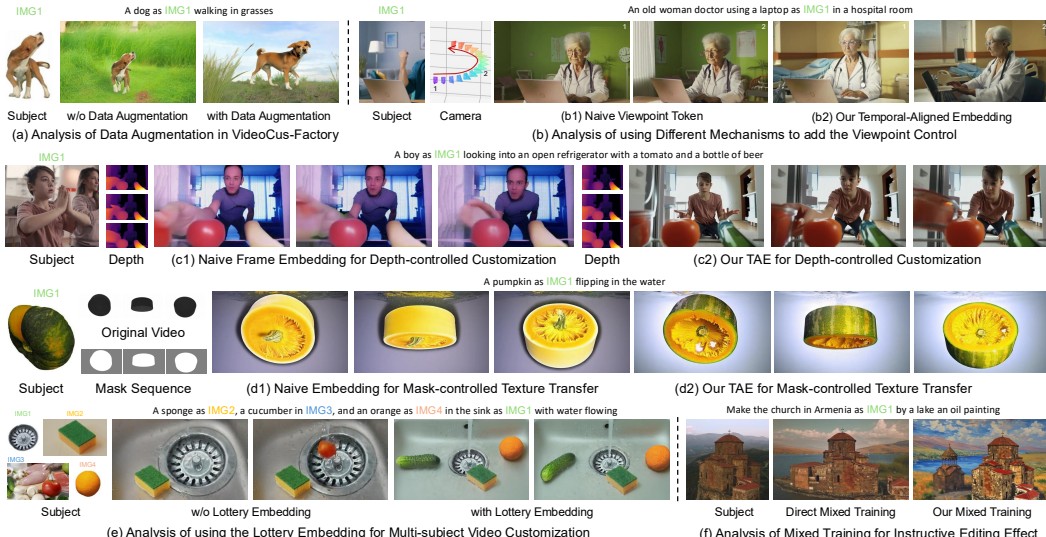

Figure 9: Visual analysis. (a) Using the data augmentation in our VideoCus-Factory can vary the scale, pose, and action of subject in the customized video. (b) Our TAE can better control the viewpoint. (c1) and (c2) show that our TAE can improve the consistency, video quality, and guidance of depth. (d1) and (d2) show that our TAE can better handle the case that the subject is not aligned with the mask by transferring the texture. (e) Using our LE better preserves the identity in zero-shot multi-subject customization. (f) studies the effect of IVTM training.

and TAE on mask-controlled customization. When the subject is unaligned with the mask of the black cylinder, the naive embedding customizes low-quality video without the pumpkin texture while generating undesired black edge. In contrast, TAE can better transfer the texture with less artifacts.

We also compare the camera embedding in TAE with the naive method that directly concatenates the viewpoint tokens into the overall long 1D tokens. This naive method leads to an increase in training time by 10% as the computational complexity of self-attention is quadratic to the length of input tokens. As shown in Fig. 9 (b1) and (b2), our TAE can control the viewpoint rotation more effectively.

**Lottery Embedding.** We conduct experiments on triple- and quadruple-subject customization that do not appear in the training data to study the effect of our LE on zero-shot more-subject customization. The averaged results are reported in Tab. 2c. When using our LE, the CLIP-T and DINO-I scores are significantly improved by 0.0623 and 0.0799. We also conduct a visual analysis in Fig. 9 (e), without using LE, the customized video misses the subject of orange and mistakenly extracts the tomato from IMG3 instead of the desired subject, the cucumber. In contrast, using our LE can accurately compose the four subjects with proper scales and physically consistent motion in the customized video.

**Image-Video Transfer Mixed Training.** We conduct experiments on the instructive editing single-subject customization in Tab. 2d to study the effect of our IVTM training strategy. CLIP-T score and the user preference percentage of IVTM training are reported. Our IVTM performs much better than no mixed training and direct mixed training with the OmniEdit dataset. We observe that the direct mixed training still can not enable some hard instructive editing categories such as removal, color change, style transfer, *etc*. Fig. 9 (f) shows an example of style transfer. The direct mixed training can not follow the editing instruction while our IVTM can customize the church in an oil painting style.

## 5 Conclusion

In this paper, we focus on studying the subject-driven video customization with different control conditions in a feedforward manner. We first propose a data construction pipeline, VideoCus-Factory, to produce training data pairs. Our VideoCus-Factory can also create depth-to-video and mask-to-video control signal data. Subsequently, we present a DiT-based framework, OmniVCus, with two embedding mechanisms, LE and TAE. LE enables more-subject video customization in inference by using limited training subjects to activate more frame position embeddings. TAE enhances the control effect of temporally aligned signals by assigning the same frame embeddings to the control tokens and noise tokens. Experiments show that our method outperforms SOTA algorithms in both quantitative and qualitative evaluations while achieving more flexible control and editing effects.

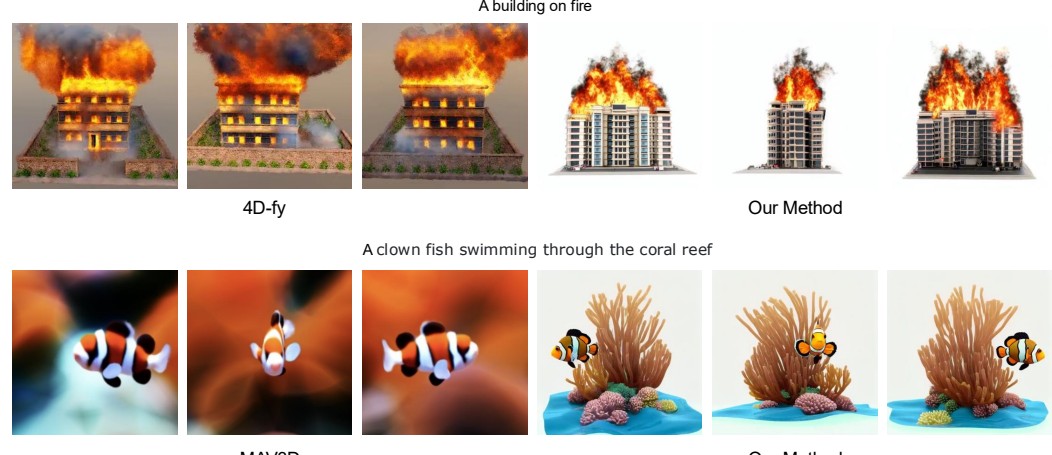

Figure 10: Text-to-4D generation comparison with 4D-fy [78] (upper) and MAV3D [79] (lower).

## 6 Limitations

The main limitation of our method is that it takes a long time (about three months) to create data. Meanwhile, the labeling quality is also constrained by the capability of the used foundational models. Specifically, the quality of the caption label and detection bounding boxes is constrained by the model capability of Kosmos-2 [61]. The quality of mask-to-video and depth-to-video control data pairs is constrained by SAM2 [62] and Video-Depth-Anything [63]. However, we believe that as the capability of these foundational models improves, the quality of data labeling will be better.

## 7 Broader Impact

Subject-driven video customization is an important and challenging topic in computer vision. It enables creators to quickly generate or edit video content based on specific subjects (people, objects or scenes), giving rise to a variety of applications: from short video clips of a few seconds on social media, to special effects synthesis at the film industry level, to personalized narratives and immersive experiences in advertising, education and digital cultural heritage. This technology not only lowers the threshold for high-quality content production, unleashes the creative potential of small and medium-sized studios and independent bloggers, but also brings higher efficiency and more flexible iteration space to the traditional film and television production process.

Until now, subject-driven video customization techniques have no negative social impact yet. Our proposed method does not present any negative foreseeable societal consequences, either.

## 8 Model Emergence Capability: Text-to-4D

As aforementioned, our model mixed trained with customization data (dynamic) and text-to-3D data (static) can perform text-to-4D generation. Please note that the 4D here refers to dynamic multi-view. Our model does not directly contain 4D representation.

We compare our method with two SOTA text-to-4D generation methods: MAV3D [79] and 4D-fy [78] in Fig. 10. Our method can generate more 3D-consistent novel views with higher dynamic degree.

Please refer to our project page for more video dynamic generation results.

## Acknowledgement

This work was supported by the office of Naval Research with award N000142412696 and in part by Adobe Inc.

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
