# OpenReview forum: "OmniVCus: Feedforward Subject-driven Video Customization with Multimodal Control Conditions"
_NeurIPS.cc/2025/Conference — NeurIPS 2025 poster_

### Official Review · Reviewer_sS6b · 2025-07-01

**Clarity:** 3
**Significance:** 3
**Originality:** 4
**Rating:** 6
**Confidence:** 5

**Summary:**

This paper presents OmniVCus, a comprehensive framework for subject-driven video customization that addresses several major limitations of prior work. The key problems it tackles are the difficulty of creating multi-subject training data and the lack of fine-grained control over the subject's actions and attributes. The authors first introduce "VideoCus-Factory," a data pipeline that automatically generates multi-subject training pairs, along with control signals like depth and masks, from unlabeled raw videos. They then propose the OmniVCus model, which uses two novel embedding techniques: "Lottery Embedding" to enable generating videos with more subjects than seen during training, and "Temporally Aligned Embedding" to better integrate control signals. Finally, they use a mixed training strategy to transfer knowledge from image-editing datasets, allowing users to edit subjects in the final video with text instructions.

**Questions:**

The "Lottery Embedding" concept for zero-shot generation of more subjects is very clever. Is there a practical upper limit to this capability? For instance, if the model is trained with a maximum of K=2 subjects, how well does the quality hold up when trying to generate M=5 or M=10 subjects at inference time?

Your "VideoCus-Factory" data pipeline is a core part of this work. How sensitive is it to potential failures from the upstream models it relies on, such as SAM-2 failing to track a subject correctly? How effectively does your subject filtering step handle this kind of noise?

The model demonstrates an impressive ability to compose multiple control signals. Have you observed any negative interference between different controls? For example, does applying a very dynamic camera motion make it more difficult for the model to also follow a precise text-based edit to a subject's clothing?

The texture transfer ability shown in Figure 1(h) and 1(i) is very interesting. Was this an emergent capability of the model, or was it explicitly trained for? It seems to prioritize the source texture over the target object's identity when the control signal is misaligned.

**Ethical Concerns:**

["NO or VERY MINOR ethics concerns only"]

**Final Justification:**

The rebuttal effectively addressed major concerns: the user study was expanded to 114 participants with strong results, subject filtering mitigates upstream model errors, and the single-stage training process is comparable to prior work with code to be released. While computational cost and some control-signal interference remain, these are acceptable limitations. The contributions remain technically solid and impactful, justifying the increased score.

**Limitations:**

Yes

**Paper Formatting Concerns:**

No major formatting issues were found.

**Quality:**

4

**Strengths And Weaknesses:**

**Strengths:**

The paper successfully tackles multiple hard problems at once: multi-subject data creation, zero-shot subject composition, and multi-modal controllability (text, mask, depth, camera).

The proposed "VideoCus-Factory" data pipeline is a major contribution, as it removes the dependency on manually labeled data for this complex task.

The novel technical ideas, particularly "Lottery Embedding" for scaling the number of subjects and "Temporally Aligned Embedding" for control, are clever and effective.

The range of capabilities demonstrated in the qualitative results is extremely impressive, showing a single model that can handle many different types of user requests.

**Weaknesses:**

The training process is very complex and computationally expensive, relying on a mix of several large datasets and significant GPU resources, which may make the results difficult to reproduce.

The data pipeline's quality is dependent on the performance of several other large pre-trained models (Kosmos-2, SAM-2), meaning errors from those models can propagate into the training data.

The user study, while a good inclusion, was conducted with a relatively small number of participants (37).

---

> ### Author Rebuttal · Authors · 2025-07-30
>
> &nbsp;
> ## Response to Reviewer sS6b
> &nbsp;
>
> `Q-1:` Questions about the complex training process
>
> `A-1:` As our goal is to achieve subject-driven video customization with multimodal control conditions in a unified framework, it is necessary and unavoidable to construct multiple large datasets with different control conditions and use sufficient GPUs to train a model with large capacity through enough iterations of training data. Other works also require complex training process with significant GPU resources. For example, FullDiT [15] requires 64 GPUs to train a DiT model in four stages by gradually adding conditions of text, camera, identity, and depth from four datasets. In contrast, our model can be trained in one stage.
>
> Besides, we will release the code for data construction and training configuration for better reproducibility.
>
> &nbsp;
> &nbsp;
>
> `Q-2:` Questions about the error propagation from the large pre-trained models to the training data
>
> `A-2:` The training data pairs for subject-driven video customization are hard to collect. Using manual annotation is time-consuming, expensive, and even unaffordable. Thus, many researchers choose to use large pre-trained models for data construction. As a result, it is difficult to completely avoid model errors. However, our data pipeline has a subject filtering process to alleviate the model error propagation. Please refer to `A-5` about the effectiveness of our subject filtering.
>
> &nbsp;
> &nbsp;
>
> `Q-3:` Small number of participants in the user study
>
> `A-3:` Following your suggestion, we increased the number of participants in the user study from 37 to 114. The updated user preference (%) of our method over other competitors are listed below:
>
> (i) Single-subject and multi-subject video customization
>
> | Method | Alignment | Identity | Quality |
> |:-:|:-:|:-:|:-:|
> | VideoBooth | 92.1 | 94.7 | 95.6 |
> | DreamVideo | 90.4 | 89.5 | 93.0 |
> | Wan2.1-I2V† | 85.1 | 87.7 | 71.9 |
> | SkyReels | 74.6 | 80.7 | 71.1 |
> | SkyReels* | 72.8 | 77.2 | 69.3 |
>
> \* = multi-subject video customization
>
> (ii) instructive editing for subject-driven video customization
>
> | Method | Alignment | Identity | Quality |
> |:-:|:-:|:-:|:-:|
> | VideoBooth | 96.5 | 93.9 | 92.1 |
> | DreamVideo | 91.2 | 98.2 | 97.4 |
> | Wan2.1-I2V | 83.3 | 84.2 | 70.2 |
> | SkyReels | 73.7 | 75.4 | 68.4 |
>
> (iii) camera-controlled subject customization
>
> | Method | Alignment | Identity | Quality |
> |:-:|:-:|:-:|:-:|
> | Motionctrl | 87.7 | 88.6 | 78.1 |
> | Cameractrl | 69.3 | 90.4 | 83.3 |
> | CamI2V | 71.9 | 82.5 | 79.8 |
>
> Our method still outperforms other competitors by a large margin. We will update the results in the revision.
>
> &nbsp;
> &nbsp;
>
> `Q-4:` The upper limit of zero-shot generation of more subjects enabled by the Lottery Embedding
>
> `A-4:` According to our observation, when $M=5$, our method can still stably insert the subjects into the customized videos but often suffers from poor harmonization performance and low physics consistency. However, our method can still generate acceptable results with a low success rate (5.3%). When $M\geq6$, our method often misses some input subjects in the customized videos, and the quality of the generated video is pretty poor. When $M=10$, our method cannot survive in this challenging setting.
>
> Thus, the upper limit for LE would be $M=5$, as you found.
>
> &nbsp;
> &nbsp;
>
> `Q-5:` How sensitive is the "VideoCus-Factory" to potential failures from the upstream models?
>
> `A-5:` We input 100 samples with varying levels of errors from upstream models into our subject filtering step. 89 of them are filtered out. All the samples where SAM-2 tracking fails have some frames where the subjects are mostly or completely unsegmented. These cases are all filtered out. Among the remaining 11 samples, 7 contain only small errors in the segmentation maps of individual frames — for example, some small parts of the subject are not segmented out. These small errors have little impact on model training. The remaining 4 samples have some frames with larger segmented background areas. However, these over-segmented errors may not be in the reference frame to affect the input images of subjects during training. Thus, the negative impacts could be smaller.
>
> &nbsp;
> &nbsp;
>
> `Q-6:` Is there any negative interference between different controls?
>
> `A-6:` Yes. But the combined control of camera motion and text-based editing of a subject's clothing performs well because these two control signals do not conflict with each other. The camera movement does not affect the clothing of the subject. As shown in Fig.4, the camera embeddings are integrated into the video noise tokens, which do not have any overlap with the text tokens.
>
> However, composing the control signals of mask and depth, mask and camera, and depth and camera may have negative interference. Firstly, the mask and depth both control the spatial structure of the generated frames. As shown in Fig.4, mask and depth share the same frame position embeddings. So, simultaneously applying these two signals may fail to control the video, e.g., the subjects are not filled in the foreground segmentation or semantically appropriate positions. In addition, although the camera signal does not have conlicts with depth and mask in model conditioning, the sequence of depth or mask maps already implicitly controls the camera movement of the generated video. Thus, simultaneously applying camera motion and depth/mask signals may also have negative interference, e.g., the camera pose motion loses its control effect.
>
> &nbsp;
> &nbsp;
>
> `Q-7:` Questions about the texture transfer ability
>
> `A-7:` The texture transfer ability is an emergent capability of our model, as we do not have any training data pairs to support texture transfer.
>
> This ability stems from our well-designed embedding mechanism and flexible framework.
>
> Firstly, our TAE assigns the same frame position embeddings to the video structure signals (depth and mask) and the video noise, while only adding the timestep embedding to the video noise to distinguish them. This key clue directs the DiT model to extract the per-frame guidance, learn the semantics, and enhance the control effects. As compared in Fig.9 (d1) and (d2), our TAE can better transfer the texture than the naive embedding.
>
> Secondly, our OmniVCus takes the concatenated tokens of all control signals as input. This flexible framework can enable some zero-shot capabilities, such as depth- or mask-controlled subject-driven video customization during inference, by directly composing different control signals. However, the mixed training data only consists of depth-to-video, mask-to-video, and subject-to-video data pairs, each containing only a subset of the control conditions, i.e., only depth, mask, or reference images. Please note that the subject and depth/mask are from different samples and do not match each other in the training data. This unpaired training strategy also allows our method to flexibly change reference images of subjects under depth or mask control signals to achieve the texture transfer effect.

---

> > ### Comment · Reviewer_sS6b · 2025-08-03
> >
> > The author's reply solved my problem, and I will improve my score.

---

> > > ### Author Response · Authors · 2025-08-03
> > >
> > > Thanks for your support and valuable comments. We really appreciate it.

---

### Official Review · Reviewer_UWzw · 2025-07-02

**Clarity:** 3
**Significance:** 1
**Originality:** 1
**Rating:** 4
**Confidence:** 3

**Summary:**

The paper presents a method for incorporating multiple different conditionings into video diffusion model, the conditioning are the following: (i) subject images from which particular objects will be extracted, (ii) pixel-aligned conditionings such as masks and depth, (iii) camera poses. For each of these 3 embeddings different conditioning mechanism is used: Lottery Embedding for (i), Temporally Aligned Embedding for (ii) and direct summation of MLP processed plucker embedding for camera.

**Questions:**

- What is the main difference of this method compared to SkyReal A2, VideoAlcmemist, VACE{1} and FullDiT{2}.
- Can the authors provide evaluation on SkyReal A2 benchmark and VACE benchmark?

{1} VACE: All-in-One Video Creation and Editing

{2} FullDiT: Multi-Task Video Generative Foundation Model with Full Attention

**Ethical Concerns:**

["NO or VERY MINOR ethics concerns only"]

**Final Justification:**

The authors provided detailed rebuttal, which partially addressed some concerns. I still believe evaluation of SkyReels-A2 in Tab. 1 is not correct. Given other reviews and detailed rebuttal, I decided to change the score.

**Limitations:**

Yes.

**Quality:**

2

**Strengths And Weaknesses:**

**Strengts:**
- The paper and method is fairly clear.
- The quantitative evaluation seems to show better results compared to new methods, such as SkyReels.

**Weaknesses:**
- The modifications proposed in this paper is quite insignificant, the data collection pipeline is quite similar to SkyReels and VideoAlchemist, mentioned in this paper. The only difference is the better filtering based on SAM2 mask tracking and data augmentation based on pasting the images onto the random background. The difference in conditioning mechanisms is also quite minor.

- It is not clear which evaluation prompts are used. It well maybe that the prompts favor the proposed method, for example prompts from the training dataset or prompts very similar to them. Why not use SkyReals-A2 benchmark or VideoAlcmemist benchmark for evalutions?

- It is not clear if the baseline models are utilized correctly. There is only one SkyReals-A2 example, and the first frame looks like a white segmented-out woman. So my guess is that segmented out frame is provided as image for i2v conditioning not as a reference image. In Fig. 6, the first frame is completely broken which also suggest that the model is used incorrectly.

- The generation quality is far from being perfect, the qualitative results of the proposed method either suffer from the copy-pasting issue (a1 last row, a3 second row first sample and forth row)  or have severe face flickering artifacts (a3 first row first sample, a3 second row second sample).

---

> ### Author Rebuttal · Authors · 2025-07-30
>
> &nbsp;
> ## Response to Reviewer UWzw
> &nbsp;
>
> `Q-1:` Comparison with concurrent works SkyReels-A2, VideoAlchemist, VACE, and FullDiT
>
> `A-1:` Our work is different in:
>
> (i) Motivation and Research Scope
>
> We propose a unified framework to comprehensively study subject-driven video customization under different control conditions including depth, mask, camera, and editing instruction. In contrast, SkyReels-A2 and VideoAlchemist only study the vanilla subject-driven customization without adding any other control signals. VACE and FullDiT are general video generation and editing frameworks. For subject-driven customization, VACE does not study camera control while FullDiT does not study mask control. In addition, three important problems in subject-driven customization are not explored by these four works: zero-shot more-subject customization, texture transfer, and instructive editing. Our work aims to fill these research gaps.
>
> (ii) Method Details
>
> Firstly, towards zero-shot more subject customization, we propose Lottery Embedding (LE). Previous methods usually assign a fixed number of frame position embeddings to the subjects and infer with the same number of subjects as used in training. Thus, when performing zero-shot more-subject customization, the additional frame embeddings have not been seen during training, making the model easily fail to insert some subjects, as shown in Fig.9 (e). To address this issue, our LE uses a limited number of subjects in the training data to activate more frame position embeddings. Thus, the additional frame embeddings have already been learned in training and can be directly assigned to more subjects in inference. As compared in Tab.2 (c), using LE can improve CLIP-T and DINO-I by 30% and 24%.
>
> Secondly, we design a TAE to better handle the control signals by leveraging their temporally aligned properties. VACE and FullDiT directly assign different frame position embeddings to mask/depth and noise. The per-frame correlation between the control signal and noise is not fully exploited. In contrast, our TAE assigns the same frame position embeddings to the control signals (depth/mask) and video noise, and adds the timestep embeddings to the noise to distinguish them. This key clue directs the model to extract the per-frame guidance. In Tab.2 (b), using TAE improves CLIP-T and DINO-I by 17% and 29%. As compared in Fig.9 (c) and (d), using TAE can better preserve the subject identity and has the emergent capability to transfer the texture when the subject is not aligned with the mask. Besides, for the camera signal, FullDiT concatenates it to the input tokens. This may lead to conflicts with depth/mask and unnecessary computational complexity, as the camera signal does not contain fine-grained information. To tackle these issues, our TAE first encodes camera poses into ray embeddings and then integrates them into noise tokens by MLPs. See Fig.9 (b) and Line 269-272, using TAE can reduce 10% training time and control the camera movement more effectively.
>
> Thirdly, to enable instructive editing effect in video customization, we design an Image Video Transfer Mixed (IVTM) training strategy. In Line 156 - 168, our IVTM builds a common task pair on image- and video-level as the bridge to transfer the instructive editing knowledge from image to video. See Tab.2 (d) and Fig.9 (f), using IVTM improves CLIP-T by 21% and enables the editing effect of style transfer, which is not supported by the four works.
>
> (iii) Comparison on data pipeline and conditioning method with SkyReels and VideoAlchemist
>
> Our data collection pipeline is different as it has subject filtering, data augmentation, and control signal construction. Our subject filtering adopts SAM2 tracking to filter out failure segmented subjects, which is very important to improve the identity preservation in model training. In data augmentation, we rescale, re-center, rotate the subject, and change its color and background to avoid the information leakage of the subject, which encourages the trained model to have more variation in customized videos. Besides, our data pipeline also constructs high-quality control signals such as depth and mask. See Tab.2 (a), using our subject filtering and data augmentation significantly improves CLIP-T and DINO-I. Besides, training with our data can yield mask-/depth-controlled effects, as shown in Fig.1.
>
> (iv) Performance Comparison
>
> We compare with FullDiT, VideoAlchemist, SkyReels, and VACE in the following table, `A-2` table, and `A-5` tables. Our method outperforms the four methods on the metrics of identity preservation (CLIP-I, DINO-I, Subj-S) and text following (CLIP-T, Text-S).
>
> |Method|CLIP-T|DINO-I|CLIP-I|Smoothness|Dynamic|Aesthetic|
> |:-:|:-:|:-:|:-:|:-:|:-:|:-:|
> |FullDiT|18.64|46.22|68.59|94.95|16.68|5.46|
> |Ours|19.72|50.06|71.83|95.30|15.29|5.61|
>
> SkyReels, VACE, and FullDiT are arxiv papers. VideoAlchemist is published by CVPR 2025, later than 1st March. According to NeurIPS policy, these four papers are concurrent works.
>
> &nbsp;
>
> `Q-2:` Questions about the evaluation prompts
>
> `A-2:` Our evaluation prompts are hand-written and fundamentally different from the training prompts generated by Kosmos-2. There is no overlap in texts, images, or videos between the testing and training sets. Besides, to ensure all baseline models perform at their best, we use their own data preprocessing pipelines to adapt the testing images and text prompts.
>
> The benchmarks of SkyReels-A2 and VideoAlcmenist only have reference images of subjects and do not support comprehensive evaluation for video customization under different control conditions such as editing instruction, camera pose, depth, and mask. In addition, their benchmarks have at most two subjects and a background for each sample. It is not sufficient for us to study zero-shot more-subject customization. Besides, fixing the third image as the background reference is also not flexible for evaluation.
>
> Thus, we collect our dataset for more comprehensive and flexible evaluation. We will also release our testing set for further research.
>
> Besides, we also test on the benchmarks of VideoAlcmemist and SkyReels-A2 in the following table and `A-5` first table. Our method performs better on the metrics of identity preservation (Subj-S, ID consistency) and text following (Text-S, Image-Text Similarity).
>
> |Method|Text-S|Vid-S|Subj-S|Dync-D|
> |:-:|:-:|:-:|:-:|:-:|
> |VideoAlchemist|0.254|0.780|0.570|0.506|
> |Ours|0.273|0.791|0.585|0.499|
>
> &nbsp;
>
> `Q-3:` Questions about how we use the baseline models
>
> `A-3:` For each baseline model, we strictly follow the instructions in its official repositories to use its code. In addition, to ensure all baseline models perform at their best, we use their own data preprocessing pipelines to adapt the testing images and text prompts to match their input requirements.
>
> SkyReels-A2 sometimes generates artifacts, flickering, cropping issues, and even corrupted early frames in customized videos when handling challenging cases. This is why the first frame of Skyreels-A2 in Fig.5 loses part of background and the first frame is broken in Fig.6. Please note that we use the original image as the reference image instead of using the segmented out image. SkyReels-A2 can only perform reference-to-video. It can not perform I2V so it does not have I2V conditioning. We have double-checked by using both inference scripts and gradio demo scripts to make sure the generation results are the same.
>
> In addition, there are also some cases (e.g., Fig.7) where SkyReels-A2 does not break early frames but still performs worse than our method.
>
> &nbsp;
>
> `Q-4:` Questions about our generation results
>
> `A-4:` (i) Although in some cases the subjects look similar in the given image and customized video, they still have some variations. In a1 last row, the face gesture and action of the boy change. The woman also changes from kneeling for picnics to standing to mop the floor. In a3 second row first sample, the man's eyes and the clothes color change. His hand also changes to hold a cup to drink coffee, while the pose of the cup also changes. In a3 forth row, the man's hand movements and clothes, and computer pose have all changed.
>
> In most cases, our customized video has a large variation on the subjects instead of simply copy-pasting them, such as a1 first row and a3 last row.
>
> (ii) a3 first row first sample and second row second sample are two challenging cases where the people's faces are very small. In the first sample, the old man's face is defocused in the reference image. As a result, his face is blurry in the customized video. In the second sample, the reference image IMG2 only shows the woman's side face. Thus, when generating the frontal face, there is a chance of producing minor flickering artifacts. However, towards the same woman subject, a2 last row shows a good customized video with her frontal face.
>
> &nbsp;
>
> `Q-5:` Evaluation on SkyReels-A2 and VACE benchmarks
>
> `A-5:` The results on SkyReels-A2 and VACE benchmarks are shown in the following tables. Our method outperforms SkyReels-A2 and VACE on the metrics of identity preservation (ID, object, and subject consistency) and text following (image-text similarity). The gaps on other metrics such as image quality mainly stem from the performance gaps between our 5B base model and Wan2.1-14B model.
>
> |Method|ID Consistency|Object Consistency|Background Consistency|Image Quality|Asthetic Quality|Motion Smoothness|Dynamic Degree|Image-Text Similarity|
> |:-:|:-:|:-:|:-:|:-:|:-:|:-:|:-:|:-:|
> |SkyReels-A2|0.398|0.809|0.677|0.683|0.579|0.891|1.000|28.188|
> |Ours|0.403|0.815|0.664|0.581|0.552|0.877|0.963|28.426|
>
> |Method|Asthetic Quality|Background Consistency|Dynamic Degree|Image Quality|Motion Smoothness|Overall Consistency|Subject Consistency|Temporal Flickering|
> |:-:|:-:|:-:|:-:|:-:|:-:|:-:|:-:|:-:|
> |VACE|63.25|98.03|30.00|72.29|99.51|25.85|98.54|99.15|
> |Ours|61.90|96.28|45.06|68.33|99.17|26.42|98.81|96.04|

---

> > ### Comment · Reviewer_UWzw · 2025-08-04
> > **Discrepancy between proposed benchmark and SkyReels-A2 benchmark**
> >
> > Thank you for the detailed response. Can you explain the large difference between the proposed benchmark in Fig. 5, and SkyReels-A2 benchmark. In Tab.5 (a) the proposed method have a huge advantage over SkyReels, while in SkyReels-A2 all the metrics are almost the same, and in some metric SkyReels is better. For example, text-similarity have a difference of 2e-3 (for SkyReels-A2), while 5e-2 (for Tab.5 (a)). Consistency and Dynamic Degree is contradicting between SkyReels-A2 and Tab.5 (a).

---

> ### Author Response · Authors · 2025-08-04
> **Discussion with Reviewer UWzw**
>
> &nbsp;
>
> Thanks for your valuable comments.
>
> I suppose you are referring to Table 1 instead of Figure 1 or Table 5, as Figure 1 does not show the quantitative comparison and we only have two tables in the main paper.
>
> Here is the detailed analysis.
>
> &nbsp;
>
> (i) The main reason for the differences in Image-Text Similarity, Consistency, and Dynamic Degree is that the prompts in the benchmark of SkyReels-A2 favor SkyReels-A2 model more. Specifically, the reference images in SkyReels-A2 testing sets follow a fixed combination: human + thing + environment. Please note that SkyReels-A2 is not limited to generating videos under this specific combination. However, since most of its training data follows this combination pattern, the model exhibits a bias toward it and performs well in such customization scenarios. In contrast, our benchmark does not always follow such fixed combination. The testing reference images do not always include a human subject or a background environment. Moreover, the positions of different input reference subjects are not fixed. In other words, our testing set considers more diverse and flexible combinations, offering better generality and enabling a more comprehensive evaluation of the model’s customization ability. However, when testing on our benchmark, we will do our best, such as changing the order of the reference subjects, to allow Skyreels to perform at its best.
>
> Therefore, on the fixed-combination benchmark that favors SkyReels-A2, the advantages of our method in ID Consistency, Object Consistency, and Image-Text Similarity are reduced. Meanwhile, our method underperforms SkyReels-A2 in terms of Background Consistency and Dynamic Degree.
>
> Besides, another key reason our method underperforms SkyReels-A2 in Background Consistency is that we do not include a background reference in every training data sample to avoid over-fitting. Instead, we adopt a more balanced foreground-background training data sampling ratio for more general and flexible customization.
>
> In addition, the Dynamic Degree metric on SkyReels-A2 is of limited reference value, as both SkyReels-A2 and Keling1.6 achieve a full score (1.000) in Table 1 of the SkyReels-A2 paper. This phenomenon also suggests, to some extent, that the testing setting of SkyReels-A2 may have some bias. In contrast, such issues do not occur in our benchmark, where all models score below 0.7 on the Dynamic Degree metric, suggesting our benchmark has stronger discriminative power and thus making the comparisons more informative and meaningful.
>
> &nbsp;
>
> (ii) The gaps in Image Quality, Asthetic Smoothness, and Motion Smoothness mainly stem from the differences in the base model's capability. Since SkyReels-A2 uses the more powerful Wan2.1-14B base model, which is superior to our internal 5B base model, it achieves better performance than our method on these three metrics.
>
> &nbsp;
>
> Anyway, SkyReels-A2 is a great work. We would like to add a detailed discussion and comparison with the SkyReels-A2 benchmark to highlight its contribution and increase its impact.
>
> &nbsp;

---

> > ### Comment · Reviewer_UWzw · 2025-08-05
> > **Reply**
> >
> > Thank you for prompt response. I see your point, however I still think that something is wrong with SkyReels-A2 evaluation in Tab.1 (a), so I encourage the authors to investigate why they are getting the broken outputs and fix it for the final version of the of the paper.
> >
> > Otherwise, my comments are adequately addressed.

---

> ### Author Response · Authors · 2025-08-06
> **Response to Reviewer UWzw**
>
> &nbsp;
>
> Thanks for your feedback.
>
> SkyReels-A2 is a great work but it does break some early frames in some challenging cases. However, it performs normally in most other cases. Not all videos generated by SkyReels-A2 exhibit this issue. We have already followed the instruction of the official repository of SkyReels-A2 to use the inference script and gradio demo script to double-check that the generation results are exactly the same.
>
> Meanwhile, we will follow your suggestion to do more sanity check. We would like to show more good cases where SkyReels-A2 does not break early frames and add more detailed explanation for SkyReels-A2 in the revision.
>
> Our testing sets will be made publicly available. Other researchers can also test SkyReels-A2 on our benchmark.
>
> Finally, we sincerely thank you for your valuable comments and active discussion with us during the rebuttal period.
>
> &nbsp;

---

### Official Review · Reviewer_oD5B · 2025-07-02

**Clarity:** 3
**Significance:** 2
**Originality:** 3
**Rating:** 4
**Confidence:** 4

**Summary:**

This paper introduces OmniVCus, a feedforward framework for subject-driven video customization that supports multi-subject scenarios and multimodal control conditions, such as depth maps, segmentation masks, camera trajectories, and textual prompts. The authors propose a data construction pipeline, VideoCus-Factory, to generate training data pairs for multi-subject customization from raw videos without requiring labeled data. The framework employs a Diffusion Transformer (DiT) architecture with two key embedding mechanisms: Lottery Embedding (LE) for enabling inference with more subjects than seen in training, and Temporally Aligned Embedding (TAE) for improved control signal integration. The method is trained using an Image-Video Transfer Mixed (IVTM) strategy, combining image editing and video customization data to enable instructive editing. Experimental results demonstrate superior performance over state-of-the-art methods in both quantitative metrics and qualitative evaluations, particularly in multi-subject customization and flexible control.

**Questions:**

1. Regarding the unintended editing effects, such as the hairstyle changes in Figure 1(g), can the authors clarify the specific control conditions or training data limitations that led to these issues?
2. The paper does not discuss the computational cost of training or inference for OmniVCus. Could the authors provide details on the training time, hardware requirements, and inference latency, particularly for multi-subject scenarios?
3. The IVTM training strategy is a key component, but its impact is only briefly analyzed in Table 2d. Could the authors provide a more detailed ablation study, including the effect of varying the ratio of image-to-video data?

**Ethical Concerns:**

["NO or VERY MINOR ethics concerns only"]

**Final Justification:**

The authors‘ responses have clarified many of the concerns I raised in the initial review.

The explanation about the misalignment between the depth map and the subject identity is reasonable and demonstrates that the model has been designed to generalize in such challenging cases. The clarification that identity preservation is better when control signals are aligned is helpful. However, it would be beneficial if future versions of the paper could emphasize this trade-off more explicitly, and perhaps suggest ways to detect or mitigate misalignment during deployment.

Considering the overall quality of the paper, I will maintain my positive score.

**Limitations:**

The authors have discussed the limitations.

**Paper Formatting Concerns:**

None.

**Quality:**

3

**Strengths And Weaknesses:**

Strengths:
* The paper presents innovations in subject-driven video customization data construction. The VideoCus-Factory pipeline is a novel approach to constructing multi-subject training data without reliance on labeled video data, addressing a critical gap in the field.
* The ability to handle multi-subject customization and incorporate diverse control conditions is a substantial advancement for real-world applications。
* The framework’s feedforward approach avoids the computational overhead of test-time tuning, making it practical for deployment.
* The paper is well-structured, with clear descriptions of the VideoCus-Factory pipeline, the OmniVCus framework, and the experimental setup.

Weaknesses:
* The fidelity of human subject customization is limited in some cases. For instance, as shown in Figure 1(g), the hairstyle of a character is altered during editing, resulting in unintended changes that deviate from the desired customization. This suggests that the model struggles with fine-grained identity preservation.
* While the paper is generally clear, some technical aspects of the OmniVCus framework, such as the precise implementation of LE and TAE, lack sufficient detail. For example, the mechanism by which LE activates additional frame embeddings for unseen subjects during inference is not fully explained.
* The paper does not adequately discuss the computational efficiency or scalability of the proposed framework. The training complexity of IVTM and the resource demands of the DiT architecture are not detailed.

---

> ### Author Rebuttal · Authors · 2025-07-30
>
> &nbsp;
> ## Response to Reviewer oD5B
> &nbsp;
>
> `Q-1:` Questions about the hairstyle changes of human subject customization.
>
> `A-1:` The change of hairstyle in Fig.1 (g) is due to the misalignment between the given subject and the depth control signal. In particular, the given subject is a woman with long hair. However, the depth sequence is estimated from a man with short hair. The depth signal controls the structure of the customized video and restricts the shape of the human subject's hair. As a result, when filling the woman into the depth of the man, our model keeps her face identity and changes her hairstyle to fit the shape of the depth map. This is a challenging case to test whether the model can handle subject and control signals that are not perfectly aligned, thereby evaluating its customization ability.
>
> Similarly, when the given subject is severely misaligned with the depth/mask control signal, our model can still transfer the texture of the subject and change its shape to fill in the semantically appropriate position, as shown in Fig.1 (h) and (i). This texture transfer effect is a feature of our well-designed frame embedding mechanism and flexible framework, as previous frame embedding method performs poorly in these challenging cases, as compared in Fig.9 (d1) and (d2).
>
> On the other hand, when the subject is well aligned with the given mask or depth, the fine-grained identity of the given subject can be better preserved. As shown in Fig.1 (f) and Fig.9 (c2), the hairstyle of the boy does not change in the customized videos because the mask and depth are all estimated from a short-haired man. In addition, when not using depth/mask control signals, the customized videos are not restricted, and the fine-grained identity of the subject can also be better preserved. As shown in Fig. 5, 6, 7, and 8, the hairstyle of the human subjects does not change because there is no mask/depth to restrict the shape of the human's hair.
>
> &nbsp;
> &nbsp;
>
> `Q-2:` Details about LE and TAE.
>
> `A-2:` We will release the code of LE and TAE for their precise implementation.
>
> (i) Additional frame embeddings for more subjects are activated during the training process. Previous methods usually assign as many frame position embeddings as the number of subjects in the training data sample. As a result, when performing zero-shot more-subject customization, the additional frame position embeddings have not been seen during training, making the model easily fail to insert some subjects, as shown in Fig.9 (e).
>
> In contrast, the core insight of our LE is to use a limited number of subjects in the training data to activate more frame position embeddings. As described in Line 127 - 137, LE randomly selects $K$ out of $M$ frame position embeddings ($M$ > $K$, $M$ = 4, $K$ = 2). Thus, these additional frame position embeddings have already been learned during the training process and can be directly assigned to more subjects in inference to enable zero-shot more-subject video customization without extra effort. As compared in Tab.2 (c), using our LE can significantly improve the CLIP-T and DINO-I scores by 30% and 24%.
>
> (ii) The technical details of TAE are described in Line 138 - 155. The key idea of TAE is to leverage the temporally aligned properties of the input conditions (depth, mask, and camera) to extract the per-frame guidance, learn the semantics, and enhance the control effects.
>
> For the depth/mask signals that control the video frame structure, TAE assigns the same frame position embeddings to them and the video noise, while only adding the timestep embedding to the noise for distinction. We notice that the camera condition does not contain fine-grained spatial and semantic information. Thus, to avoid conflicts with depth/mask and reduce the computational complexity, TAE adopts the Plücker coordinates to encode the camera poses into pixel-aligned ray embeddings and then integrates these ray embeddings into video noise by MLPs.
>
> As shown in Tab.2 (b), Fig.9 (b), (c), (d), and Line 269-272, using TAE can better preserve subject identity, reduce training time, enhance control effect, and enable the model's emergent capability of texture transfer.
>
> &nbsp;
> &nbsp;
>
> `Q-3:` Training time, hardware requirements, inference latency, and scalability
>
> `A-3:` (i) As described in Line 191, we use 64 A100 GPUs to train our OmniVCus for 5 days.
>
> (ii) The inference time (m = minutes, s = seconds) for different tasks on a single A100 GPU at a spatial resolution of 384$\times$640 for 64 frames at 24 fps are shown in the following table:
>
> | Num of Subjects | Only Subject-driven | Instructive Editing | Camera-controlled | Depth-controlled | Mask-controlled |
> |:-:|:-:|:-:|:-:|:-:|:-:|
> | 1 | 3m 51s | 3m 51s | 3m 53s | 7m 11s | 7m 11s |
> | 2 | 3m 57s | 3m 57s | 3m 59s | 7m 23s | 7m 23s |
> | 3 | 4m 4s | 4m 4s | 4m 7s | 7m 36s | 7m 36s |
> | 4 | 4m 11s | 4m 11s | 4m 14s | 7m 49s | 7m 49s |
>
> (iii) We show the ablation study about the scalability of batch size and spatial resolution on the single-subject video customization task in the following two tables.
>
> | Batch Size | 89 | 178 | 267 | 356 |
> |:-:|:-:|:-:|:-:|:-:|
> | CLIP-T | 0.2726 | 0.3008 | 0.3145 | 0.3293 |
> | DINO-I | 0.4481 | 0.4830 | 0.5027 | 0.5215 |
> | Consistency | 0.9535 | 0.9867 | 0.9913 | 0.9928 |
> | Dynamic | 0.4794 | 0.5206 | 0.5563 | 0.5541 |
>
> | Spatial Size | 96$\times$160 | 192$\times$320 | 288$\times$480 | 384$\times$640 |
> |:-:|:-:|:-:|:-:|:-:|
> | CLIP-T | 0.3017 | 0.3169 | 0.3240 | 0.3293 |
> | DINO-I | 0.4985 | 0.5087 | 0.5157 | 0.5215 |
> | Consistency | 0.9892 | 0.9910 | 0.9922 | 0.9928 |
> | Dynamic | 0.5428 | 0.5693 | 0.5516 | 0.5541 |
>
> When we fix the spatial size at 384$\times$640 and increase the batch size, or fix the batch size at 356 and increase the spatial size, the video generation quality is gradually improved.
>
> &nbsp;
> &nbsp;
>
> `Q-4:` More detailed ablation study of varying the ratio of image-to-video data
>
> `A-4:` The ablation study of varying the training data ratio between the task pair:  single-subject image customization and single-subject video customization is shown in the following table. IVTM yields the best instructive editing performance (the highest CLIP-T score) when the ratio of image-to-video data is 4:1.
>
> | Image : Video | 2:1 | 3:1 | 4:1 | 5:1 | 6:1 |
> |:-:|:-:|:-:|:-:|:-:|:-:|
> | CLIP-T | 0.2714 | 0.2955 | 0.3126 | 0.3007 | 0.3082 |

---

> > ### Comment · Reviewer_oD5B · 2025-08-06
> >
> > I appreciate the authors‘ responses, which have clarified many of the concerns I raised in the initial review.
> >
> > The explanation about the misalignment between the depth map and the subject identity is reasonable and demonstrates that the model has been designed to generalize in such challenging cases. The clarification that identity preservation is better when control signals are aligned is helpful. However, it would be beneficial if future versions of the paper could emphasize this trade-off more explicitly, and perhaps suggest ways to detect or mitigate misalignment during deployment.

---

> > > ### Author Response · Authors · 2025-08-06
> > > **Response to Reviewer oD5B**
> > >
> > > &nbsp;
> > >
> > > Thanks for your support and suggestion.
> > >
> > > You are right. We will follow your advice to add more detailed explanation to explicitly emphasize the trade-off in the revision and suggest a detection- or segmentation-based method to mitigate misalignment in application.
> > >
> > > &nbsp;

---

### Official Review · Reviewer_uLej · 2025-07-02

**Clarity:** 4
**Significance:** 3
**Originality:** 3
**Rating:** 5
**Confidence:** 3

**Summary:**

This paper addresses the limitations of existing feedforward subject-driven video customization methods, which struggle with multi-subject scenarios and lack fine-grained control. First, the authors propose a data pipeline to create multi-subject training data with control signals, and an IVTM training strategy to enable instructive editing capability transferred from image-only data. The proposed model, OmniVCus, equipped with Lottery Embedding and Temporally Aligned Embedding, extends to more subjects and improves control alignment.

**Questions:**

* The Image-Video Transfer Mixed Training section is somewhat unclear. A more detailed explanation would help clarify how the image-level instructive editing knowledge is transferred to the video domain.
* How many sampling steps are used during inference for both the proposed method and the baselines? Also, are all methods built upon the same foundation model to ensure a fair comparison?
* Why is text-to-multiview data used, as mentioned in Line 184? Its role is not clearly explained in the paper.
* Additional suggestion: There is a relevant work [1] on tuning-based multi-subject video customization. Although a direct comparison may not be feasible due to the lack of released code, it is still suggested to cite it in the related work to help strengthen the completeness of the literature review.

[1] Huang, Chi-Pin, et al. "Videomage: Multi-subject and motion customization of text-to-video diffusion models." Proceedings of the Computer Vision and Pattern Recognition Conference. 2025.

**Ethical Concerns:**

["NO or VERY MINOR ethics concerns only"]

**Final Justification:**

After reading the authors' rebuttal, I've decided to keep my score at 5. Their explanations of Lottery Embedding and Temporally Aligned Embedding are clear and make sense, especially in addressing the scalability challenges of multi-subject customization and improving control effects. The clarification of the Image-Video Transfer Mixed (IVTM) training also helped me better understand how the model transfers editing ability from image to video tasks.

While the method itself is relatively simple and doesn't involve particularly novel techniques, the overall design is thoughtful and works well in practice. The authors also give clear answers regarding sampling steps, model differences, and the role of text-to-multiview data. The paper is overall good, and the response helped confirm the strength and usefulness of the proposed approach.

**Limitations:**

yes

**Quality:**

4

**Strengths And Weaknesses:**

**Strengths**

The setting of the paper is novel. It extends beyond prior work that can only achieve text+image to video generation by incorporating diverse control signals and enabling the instructive editing capability. This method expands the flexibility and applicability of subject-driven video customization. The project page demo provided in the supplementary material is impressive, especially the illustrations of emergent Text-to-4D capability. The paper is well-written and easy to follow. The conducted experiment is solid, with comparisons against state-of-the-art baselines on subject customization, instructive editing, and camera-controlled tasks, respectively.

**Weaknesses**

The proposed method is relatively straightforward, and the ideas of Lottery Embedding and Temporally Aligned Embedding are not particularly novel. Additionally, the Image-Video Transfer Mixed Training section is a little bit unclear and would benefit from a more detailed explanation.

---

> ### Author Rebuttal · Authors · 2025-07-30
>
> &nbsp;
> ## Response to Reviewer uLej
> &nbsp;
>
> `Q-1:` Questions about the Lottery Embedding (LE) and Temporally Aligned Embedding (TAE)
>
> `A-1:` (i) The motivation of designing LE is to enable zero-shot more-subject customization in inference while using training samples with limited subjects. This is a very important yet less explored problem, as constructing multi-subject-to-video training data pairs is very challenging and hard to scale up. Previous methods usually assign a fixed number of frame position embeddings to the subjects and infer with the same number of subjects as used during training. As a result, when performing zero-shot more-subject customization, the additional frame embeddings have not been seen during training, making the model easily fail to insert some subjects, as shown in Fig. 9 (e).
>
> To address this issue, our LE uses a limited number of subjects to randomly activate more frame embeddings in training. As these additional embeddings have already been learned in training, they can be directly assigned to more subjects without extra efforts in inference to enable zero-shot more-subject video customization. As shown in Tab.2 (c), using our LE can significantly improve the CLIP-T and DINO-I scores by 30% and 24%. As compared in Fig.9 (e), using our LE can better preserve the subjects in the customized videos than the naive embeddings used by previous methods.
>
> (ii) Our TAE is designed to better handle the control signals of depth, mask, and camera by leveraging their properties that are temporally aligned with the video noise. Previous methods usually indiscriminately concatenate the tokens of all control signals (depth and camera) and assign different frame position embeddings to the control signals and the video noise. This naive embedding scheme does not fully exploit the per-frame correlation between the control signal and video noise to guide the video customization. Besides, concatenating camera poses may easily result in conflicts with depth/mask to degrade control effect and unnecessary computational complexity, as the camera signal does not contain fine-grained information.
>
> To cope with these problems, TAE assigns the same frame position embeddings to the video structure control signals (depth/mask) and the video noise, and only adds the timestep embeddings to the video noise to distinguish them. This key clue directs the model to extract the per-frame guidance, learn the semantics, and enhance the control effects. As listed in Tab.2 (b), using TAE can significantly improve the CLIP-T and DINO-I scores by 17% and 29%. As compared in Fig.9 (c1), (c2), (d1), and (d2), using TAE can better preserve the identity of the subject than the naive embedding scheme and even has the emergent capability to transfer the texture when the subject is not aligned with the mask.
>
> Besides, we observe that the camera signals do not contain fine-grained spatial and semantic information. Thus, to avoid the conflicts with depth/mask and reduce the computational complexity, our TAE first adopts the Plücker coordinates to encode the camera poses into pixel-aligned ray embeddings and then integrates these ray embeddings into noise tokens by MLPs. As shown in Line 269-272 and Fig.9 (b), using our TAE can reduce 10% training time and control the camera movement more effectively.
>
> &nbsp;
> &nbsp;
>
> `Q-2:` Explanation of the Image-Video Transfer Mixed (IVTM) Training
>
> `A-2:` As analyzed in Line 156 - 168, the core idea of our IVTM training is to build a common task pair on image-level and video-level as the bridge to transfer the instructive editing knowledge from image to video. In particular, we set this common task pair as single-subject image and video customization. We choose this task pair because single-subject image customization has the same input/output format (image-to-image) with image instructive editing, and single-subject video customization has the same input/output format (image-to-video) with instructive editing subject-driven video customization. Benefit from the same input/output format, mixed training this task pair with image instructive editing dataset can transfer the knowledge and enable zero-shot instructive editing subject-driven video customization.
>
> To better transfer the knowledge, we align the frame position embeddings, as shown in the green small cubes in Fig.4, of the input images of the task pair and image instructive editing.
>
> To enable the instructive editing effect in multi-subject video customization, we assign the above frame position embeddings by our Lottery Embedding mechanism in Eq.(1) with $K = 1$, also shown in Fig.4 (a). Through our LE mechanism, more frame position embeddings are activated with the instructive editing knowledge transfer during the training process. Thus, we can directly impose editing instruction on multiple subjects in inference, as shown in Fig. 2 (i).
>
> &nbsp;
> &nbsp;
>
> `Q-3:` Questions about the inference sampling steps and foundation model.
>
> `A-3:` The sampling steps are listed in the following table.
>
> | Method | VideoBooth | DreamVideo | Wan2.1-I2V | SkyReels | Motionctrl | Cameractrl | CamI2V | Ours |
> |:-:|:-:|:-:|:-:|:-:|:-:|:-:|:-:|:-:|
> | Sampling Steps | 250 | 50 | 50 | 50 | 25 | 25 | 25 | 50 |
> | foundation model | Stable-Diffusion-v1.4-1B | Stable-Diffusion-v1.5-1.4B | Wan2.1-14B | Wan2.1-14B | Stable-Video-Diffusion-1.5B | Stable-Video-Diffusion-xt-2.3B | DynamiCrafter-1.4B | Internal-5B |
>
> Different methods are built upon different foundation models, as shown in the above table. Because the training code and data of different methods are not all publicly available, it is not feasible to implement different methods on the same foundation model.
>
> However, compared to the SOTA methods, SkyReels and Wan2.1-I2V equipped with OmniGen, our method uses a smaller model (5B vs 14B) but yields better customization results and more unified control effects under multi-modal conditions, as shown in Fig. 1, 5, 6, and 7.
>
> &nbsp;
> &nbsp;
>
> `Q-4:` Why using the text-to-multiview data?
>
> `A-4:` The text-to-multiview dataset provides camera-to-video data pairs. We mixed train our model with the text-to-multiview data to enable camera-controlled subject-driven video customization. As shown in Fig.4, the text-to-3D dataset provides control conditions of camera poses, and some other datasets provide the control signals of reference images of subjects during training. Although these control conditions are not matched to the same sample in the training process, our method can flexibly compose them (texts, camera poses, and images of subjects) in inference to enable camera-controlled subject-driven video customization, as shown in Fig.1 (d), Fig.2 (ii), and Fig.8
>
> &nbsp;
> &nbsp;
>
> `Q-5:` Cite the relevant work VideoMage
>
> `A-5:` Thanks for reminding us. VideoMage is a great work. We will not only cite it but also discuss it in detail to highlight its contribution and increase its impact.

---

### Decision · Program_Chairs · 2025-09-17

**Decision:**

Accept (poster)

**Comment:**

This paper deals with the challenging multiple-subject video customization task with versatile control conditions. It brings new contributions in regard to dataset construction and effective embedding mechanisms, leading to superior performance in quantitative and qualitative evaluations. Reviewers are positive on the paper's novelties, clarity, and performance gain. Yet, reviewer oD5B raised a concern related to alignment that might affect usage in practice, and reviewer UWzw wondered about the potential unfair comparison with SkyReels-A2. In spite of that, both reviewers rated Weak Accept, and the authors promised to carefully improve the final paper to address these concerns. On the basis of the author–reviewer discussion and the reviewers' recommendations, the AC believes this paper deserves acceptance as a poster.